# ALBI Score Is a Strong Predictor of Toxicity Following SIRT for Hepatocellular Carcinoma

**DOI:** 10.3390/cancers13153794

**Published:** 2021-07-28

**Authors:** Céline Lescure, Florian Estrade, Maud Pedrono, Boris Campillo-Gimenez, Samuel Le Sourd, Marc Pracht, Xavier Palard, Héloïse Bourien, Léa Muzellec, Thomas Uguen, Yan Rolland, Etienne Garin, Julien Edeline

**Affiliations:** 1Centre Eugène Marquis, Medical Oncology, 35043 Rennes, France; c.lescure@rennes.unicancer.fr (C.L.); f.estrade@rennes.unicancer.fr (F.E.); m.pedrono@ch-bretagne-sud.fr (M.P.); s.lesourd@rennes.unicancer.fr (S.L.S.); m.pracht@rennes.unicancer.fr (M.P.); h.bourien@rennes.unicancer.fr (H.B.); l.muzellec@rennes.unicancer.fr (L.M.); 2Centre Eugène Marquis, Clinical Research, 35043 Rennes, France; b.campillo@rennes.unicancer.fr; 3Centre Eugène Marquis, Nuclear Medicine, 35043 Rennes, France; x.palard@rennes.unicancer.fr (X.P.); e.garin@rennes.unicancer.fr (E.G.); 4CHU Pontchaillou, Hepatology, 35043 Rennes, France; thomas.uguen@chu-rennes.fr; 5Centre Eugène Marquis, Interventional Radiology, 35043 Rennes, France; y.rolland@rennes.unicancer.fr

**Keywords:** radioembolization, Yttrium-90, liver function, toxicity

## Abstract

**Simple Summary:**

SIRT, also known as radioembolization, is a new treatment for hepatocellular carcinoma. However, the precise role is still not clearly determined. Here, we describe how a new score, ALBI, used to better assess liver functions, can help to select patients for this treatment. We clearly showed that the ALBI grade was associated with toxicity and survival after this treatment.

**Abstract:**

Background: Selective internal radiation therapy (SIRT) is an innovative treatment of hepatocellular carcinoma (HCC). The albumin-bilirubin (ALBI) score was designed to better evaluate liver functions in HCC. Methods: We studied, retrospectively, data from patients treated with SIRT for HCC. The primary endpoint was the occurrence of radioembolization-induced liver disease (REILD). The secondary endpoint was overall survival (OS). Results: 222 patients were studied. The ALBI grade 1 patients had significantly less REILD (3.4%) after the first SIRT than ALBI grade 2 or 3 patients (16.8%, *p* = 0.002). Of the 207 patients with data, 77 (37.2%) had a worsening of ALBI grade after one SIRT. The baseline ALBI grade was significantly associated with OS (*p* = 0.001), also in the multivariable analysis. The ALBI grade after the first SIRT was significantly associated with OS (*p* ≤ 0.001), with median OS of 26.4 months (CI 95% 18.2–34.7) for ALBI grade 1 patients (*n* = 48) versus 17.3 months (CI 95% 12.9–21.8) for ALBI grade 2 patients (*n* = 123) and 8.1 months (CI 95% 4.1–12.1) for ALBI grade 3 patients (*n* = 36). Conclusions: The baseline ALBI grade is a strong predictor of REILD. The baseline ALBI score and variations of ALBI are prognostic after SIRT.

## 1. Introduction

Hepatocellular carcinoma (HCC) is the seventh commonest cancer and the fourth commonest cause of cancer-associated death worldwide in 2017 [1]. Staging systems include tumor burden, performance status (PS) and liver function [2,3]. The Child−Pugh (C-P) scoring system has been widely used to measure liver function [4,5]. The albumin-bilirubin (ALBI) grade was recently proposed as an objective alternative to the C−P classification [6]. This model demonstrated its usefulness for the prognosis evaluation of HCC patients who underwent liver resection for early stage disease or patients who were treated with sorafenib in advanced stage disease [7,8] but its role in clinical decision making or stratification in research trials is not yet clearly defined.

Selective internal radiation therapy (SIRT) using Yttrium-90 microspheres was studied for the treatment of HCC, especially in patients with portal vein thrombosis [9]. The aim of SIRT is to deliver a tumoricidal dose to the tumor while sparing nontumoral liver tissue [10]. Current data show a good safety profile and local tumor control, but phase 3 trials failed to show overall survival (OS) benefit compared to sorafenib in BCLC-B and -C patients [11,12,13]. These trials, however, did not use optimal dosimetry, as was shown in a recent randomized phase 2 study that demonstrated that personalized dosimetry was associated with a higher response rate and overall survival [14,15]. The subgroup of patients benefitting from SIRT still needs to be defined. The specific liver toxicity of SIRT is the radioembolization-induced liver disease (REILD). The ALBI grade was previously studied in the context of SIRT, as a prognostic factor [16,17,18,19]. However, no previous study linked the ALBI grade with REILD, either as a predictor of occurrence or to describe the evolution of REILD. Furthermore, the description of the evolution of liver function is understudied and might be improved with ALBI.

The aim of the study was to demonstrate the value of ALBI as a predictor of toxicity following SIRT, focusing on REILD, and to study the variations of ALBI after SIRT. We also wanted to confirm the prognostic value of ALBI, and to study the variation of ALBI after SIRT as a prognostic factor.

## 2. Materials and Methods

### 2.1. Study Design

This retrospective and noninterventional cohort study included HCC patients undergoing SIRT at our center from January 2008 to January 2017. They could have previously been treated or not. They all met the following inclusion criteria: age ≥18 years old, PS ≤ 2, treatable by SIRT therapy. This study received institutional review board approval.

### 2.2. Therapy Protocol

Pretreatment planning angiography and technetium-99m macroaggregated albumin scans were performed to predict the microsphere distribution and identify potential lung or digestive shunt [20]. In our cohort, SIRT was mostly performed with glass-based microspheres. The dosimetry aimed at providing 80–150 Gy to the perfused volume. In selected patients, the dose to the perfused volume could exceed 150 Gy if the mean dose to the whole liver was below 150 Gy, following the concept of personalized dosimetry [14,21]. The activity of resin microspheres was calculated using body surface area (BSA) method.

### 2.3. Assessments

Treatment efficacy was assessed by liver MRI or CT-scan by a radiologist using the mRECIST criteria. Biological and clinical follow-up were regularly performed and consisted of a clinical exam, an interview looking for potential adverse effects, and blood tests including complete blood counts, liver function tests, and kidney functions.

Baseline assessment was made within 2 weeks before pretreatment angiography, then follow-up assessments were done between 6 weeks and 3 months after first SIRT or second SIRT and then regularly every 3 to 4 months until progression or death. The variation of ALBI score was assessed at the time of first imaging evaluation. Adverse events were graded according to NCI-CTCAE v5.0. REILD was defined according to previous publications [22,23] as the appearance of a serum total bilirubin of 51.3 µmol/L or higher and ascites (clinically or by imaging) within 3 months following SIRT in the absence of tumor progression or bile duct obstruction. Death during the first 3 months were retrospectively adjudicated as related to disease progression or toxicity of SIRT according to the following principle in case of liver dysfunction: if an evaluation was performed before death, and progression was seen, liver dysfunction was deemed related to tumor progression; if an evaluation was done without progression, liver dysfunction was deemed related to SIRT; if no evaluation was done, liver dysfunction was considered as from an indeterminate cause.

### 2.4. Calculation of Liver Function Scores

The CP score was then calculated from raw data, according to the original publication and adaptation for normalization of units used (Appendix A). If the patient was recorded as receiving anticoagulation treatment, the CP coagulation score was assumed to be 1. The ALBI score was calculated using the formula: ALBI score = (log10 bilirubin × 0.66) + (albumin × −0.085), and grades were attributed as follows: grade 1 if score ≤ −2.60; grade 2 if score > −2.60 but ≤ −1.39; grade 3 if score > −1.39 [6].

### 2.5. Statistical Analysis

*p*-values for toxicity comparison according to the different ALBI grades was calculated using a Chi2 test when the groups were made of at least 5 patients and the Fisher’s exact test when they were not. Logistic regression analysis was performed to assess the odd ratio of occurrence of REILD according to ALBI grade. Kaplan−Meier curves were constructed for OS, and log-rank tests were performed to compare survival distributions. Overall survival was defined as the time between first SIRT and death or last follow-up. Univariate and multivariate Cox regression analysis was performed to calculate unadjusted hazard ratios (HR) and 95% confidence intervals (CI). Variables statistically significant (*p* < 0.1) were entered in the multivariate model with backward selection. Statistical analysis was performed with SPSS v18. All reported *p* values were two-sided and considered statistically significant when <0.05.

## 3. Results

### 3.1. Patient Characteristics

From January 2008 to January 2017, 222 HCC patients underwent SIRT at our center. The clinical characteristics of the patients included in the cohort and the treatment received are presented in Table 1. Median follow-up was 63 months.

### 3.2. Prognostic Value of Baseline ALBI Score

Log-rank test and univariable Cox proportional hazards models indicated that baseline ALBI grade was significantly associated with OS (*p* = 0.001), with a median OS of 24.0 months (95% CI 20.7–27.4) for patients with ALBI grade 1 (*n* = 88) versus 12.9 months (95% CI 9.5–16.4) for ALBI grade 2 (*n* = 130) (*p* = 0.001 for ALBI 1 vs. 2) versus 8.3 months (95% CI NE–18.8) for ALBI grade 3 (*n* = 4) (*p* = 0.190 for ALBI 2 vs. 3) (Figure 1A).

The univariable Cox-regression model analysis of variables potentially associated with OS is presented in Table 2. C-P score, α-fetoprotein (AFP), macrovascular invasion and ALBI (either score or grade, with patients with ALBI grade 2 and 3 pooled due to limited numbers of ALBI grade 3 patients) were associated with OS on univariable analysis.

Upon multivariable analysis including either ALBI score as a continuous variable or as a 3- point grade, only macrovascular invasion (*p* = 0.059 and *p* = 0.026 respectively) and baseline ALBI grade (*p* < 0.001 and *p* < 0.001 respectively) were found to be independent prognostic factors for survival after SIRT. In this cohort, ALBI score or grade were the stronger prognostic factors.

In patients classified as C−P score A5 (*n* = 131), ALBI grade 1 patients (*n* = 79) had significantly better survival than ALBI grade 2 (*n* = 52), with median OS of 24.2 (95% CI 21.0–27.4) and 11.0 months (95% CI 6.8–15.1; *p*= 0.002), respectively (Figure 1B). The difference was not statistically significant in C−P score A6 (*n* = 70, 9 ALBI grade 1 and 61 grade 2) (*p* = 0.493) and C−P B7 (*n* = 18) (*p* = 0.089) patients, probably because of the small numbers in these subgroups.

### 3.3. Baseline ALBI Grade Was Strongly Associated with Toxicity, Including REILD, after First SIRT

Toxicity follow-up was available for 218 patients. Patients with baseline ALBI grade 1 had significantly less hospitalization, grade 3/4 adverse events, ascites, hospitalization and REILD three months after first SIRT than patients with ALBI grade 2 or 3 (Figure 2). The odds ratio for the development of REILD for ALBI grade 2 or 3 vs. grade 1 patients was 5.7 (95%CI: 1.6–19.5). There was also a trend for death during the first 3 months (related or not to the treatment). Baseline ALBI grade 1 patients also showed a trend for less REILD after second SIRT (2/26 = 8%) than ALBI grade 2 or 3 patients (8/28 = 29%, *p* = 0.079).

Appendix A shows all causes of death within three months after SIRT. Most of them were found to be secondary to liver decompensation due to cancer progression or SIRT toxicity. The ALBI score seemed more discriminating than the C−P score as an initial selection criterion, since all patients who died within three months after SIRT were C−P A patients while only one patient was ALBI grade 1, this patient having experienced esophageal varices bleeding, probably not directly related to SIRT.

### 3.4. ALBI Score Variation Following SIRT

Two hundred and seven patients had ALBI evaluation after SIRT. The ALBI score frequently increased after SIRT, showing a worsening of liver function. It increased by a median of 0.35 points (standard deviation (SD) = 0.45) after one SIRT and 0.55 points (SD = 0.57) after two SIRT. Of the 207, 77 (37.2%) patients had a worsening of ALBI grade after 1 SIRT, and a further 20 out of 51 (39.2%) had a worsening after the second SIRT (Figure 3). Only 46% of patients with baseline grade 1 remained grade 1 after 1 SIRT, and only 33% of them remained so after the second SIRT.

The ALBI grade after the first treatment was also significantly associated with OS with a median OS of 26.4 months (CI 95% 18.2–34.7) for ALBI grade 1 patients (*n* = 48) versus 17.3 months (CI 95% 12.9–21.8) for ALBI grade 2 patients (*n* = 123) and 8.1 months (CI 95% 4.1–12.1) for ALBI grade 3 patients (*n* = 36) (Figure 3A, *p* < 0.001 overall). The results were significant when considering ALBI grade 1 versus 2 (*p* = 0.003) and for ALBI grade 2 versus 3 (*p* = 0.001).

The ALBI grade after the second treatment was significantly associated with OS, but the analysis was limited by the low number of patients. The median OS for ALBI grade 1 patients (*n* = 8) was 25.3 months (CI 95% 10.9–39.7) versus 18.9 months (CI 95% 11.4–26.5) for ALBI grade 2 patients (*n* = 33) and 7.1 months (CI 95% 5.4–8.8) for ALBI grade 3 patients (*n* = 10) (Figure 3B *p* < 0.001 overall). Comparison of ALBI grade 2 versus 3 was significant (*p* = 0.03), but comparison of ALBI grade 1 vs. 2 was limited by low numbers (*p* = 0.23).

In a multivariable analysis, variations of ALBI score after SIRT was also significantly associated with OS. Continuous ALBI score variation remained a significant independent prognostic factor for survival together with macrovascular invasion (HR 2.08 (95% CI 1.46–2.98), *p* < 0.001).

## 4. Discussion

In this study, several important properties of the use of ALBI grade in the context of SIRT were shown. The baseline grade was for the first time clearly and strongly associated with toxicity, including REILD. ALBI score was confirmed as an important baseline prognostic factor. Moreover, its variations were studied for the first time, and two important messages arise: ALBI score variations are frequent after SIRT and are also an independent prognostic factor for OS.

This study demonstrated that severe toxicities were far less frequently seen in patients with ALBI grade 1 patients. Adverse events grade 3 or more were six times less frequently seen in this population. Specifically, the occurrence of REILD was much lower in ALBI grade 1 patients (3.4%) vs. grade 2 or 3 (16.8%), which could clearly help to assess risk/benefit ratio for treatment. Moreover, there was a strong trend for a prediction of death during the first 3 months. Conversely C−P classification was not able to predict for death following treatment, with all patients dying of liver dysfunction within 3 months classified as C−P A, all but one of these patients were classified as ALBI grade 2. This relationship between ALBI and toxicity could help with patient selection for treatment. Specifically, the very high rates of severe toxicity of ALBI grade 3 patients should lead to contraindicate the treatment in them. However, it should also guide clinicians for the follow-up after treatment, patients with ALBI grade 2 probably needing more frequent clinical and liver function follow-up assessments after SIRT than patients with ALBI grade 1. The first randomized studies to compare SIRT with sorafenib in advanced HCC, the SARAH, SORAMIC and SIRVENIB trials were negative [11,13,24], but this result might be related to flaws in the trial designs [13]. Indeed, it seems essential to consider the tumor-absorbed dose and liver-absorbed dose to assess efficacy and potential toxicities of SIRT [14,25]. Patient selection could also be improved by better evaluation of liver function with ALBI.

Moreover, ALBI score and grade frequently increased after SIRT. Faced with the multiplication of systemic therapeutic possibilities, it is essential to monitor the evolution of liver function and to keep a good hepatic function to offer as many therapeutic lines as possible to the patient, possibly switching earlier to systemic treatment when liver function tends to deteriorate under locoregional therapies [26]. The combination of SIRT with systemic therapies should also be studied.

Our data confirmed that ALBI grade at baseline is better than C−P score for the discrimination of the prognosis of HCC patients treated with SIRT [16,17]. The baseline ALBI grade enabled us to divide patients into three groups characterized by statistically and clinically different median OS (baseline ALBI grade 1: 23.9 months, baseline ALBI grade 2: 12.9 months, baseline ALBI grade 3: 8.3 months). These results confirm those found by Hickey et al., whose population had baseline worse liver functions [16]. In our population with better liver function, ALBI was still better than C−P for defining prognostic groups, and, taken together with our data on toxicity, this clearly indicated that ALBI should be used as a stratification factor in prospective trials.

Our study has some limitations. First, the size of our cohort was not big enough to find a significant difference of median survival among C−P score A6 or C−P score B, contrary to Antkowiak et al. [17]. Secondly, it was based on a single-center population, which may occasion a selection bias. In addition, some patients had intra-arterial treatment before SIRT, which probably changes the hepatic tolerance of SIRT. Importantly, with a single-arm study, we could not compare evolution of ALBI score after SIRT with evolution of ALBI score under natural progression of HCC, under a natural decrease of liver function due to cirrhosis, or under alternative treatment. Finally, its retrospective character warrants assessment in prospective clinical trials.

## 5. Conclusions

Baseline ALBI could be a useful tool for the selection of patients for SIRT. ALBI can predict the risk of occurrence of toxicity. Moreover, its variations also have important prognostic information, and deterioration of ALBI score is frequent after SIRT. In view of the multiplication of systemic treatments, it is important not to deteriorate the liver functions of patients with local treatment, and deterioration of ALBI score might be an important parameter to consider in this context. Hence, it would be interesting to retrospectively study the ALBI score and its variations in previous phase III trials. In addition, they should be taken into account for the next studies to come.

## Figures and Tables

**Figure 1 cancers-13-03794-f001:**
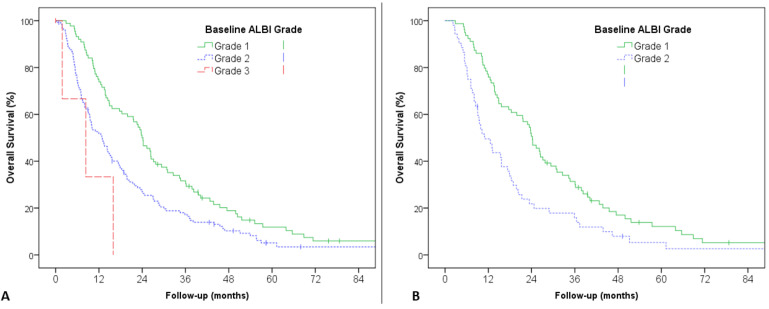
Overall survival according to baseline ALBI grade, (**A**) in the overall population, and (**B**) in patients with Child−Pugh A5.

**Figure 2 cancers-13-03794-f002:**
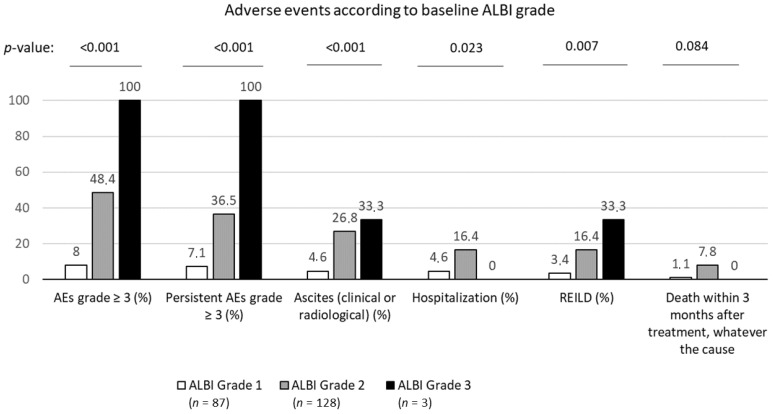
Occurrence of main toxicities, according to baseline ALBI grade. AEs: adverse events; REILD: radioembolization-induced liver disease. *p*-values were calculated by Chi-square test for trend.

**Figure 3 cancers-13-03794-f003:**
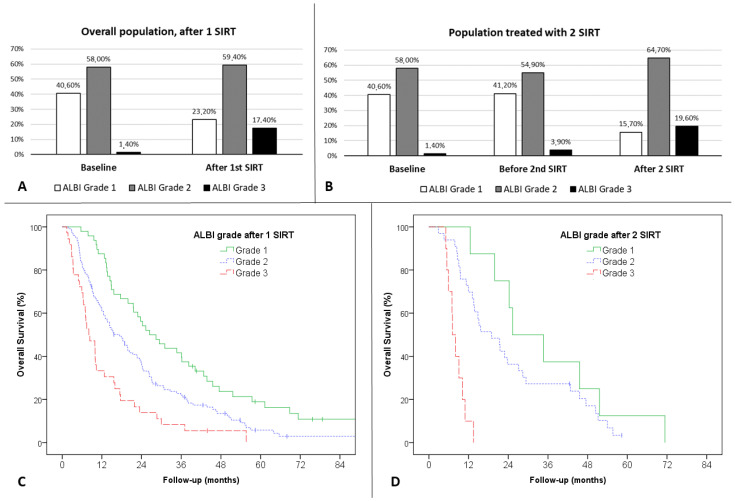
ALBI grade variations after treatment: (**A**) proportion of patients in each ALBI grade, before and after SIRT after 1 SIRT; (**B**) proportion of patients in each ALBI grade, before and after 1 and 2 SIRT, for patients treated with 2 SIRT; (**C**) overall survival according to ALBI grade after first treatment; (**D**) overall survival according to ALBI grade after two treatments.

**Table 1 cancers-13-03794-t001:** Patients, tumors and treatment characteristics.

Characteristics		*n* = 222
**Age, median**		66.4 ± 8.9
Gender	Female	28 (12.6%)
	Male	194 (87.4%)
Performance Status	0	149 (67.1%)
	1 or 2	73 (32.9%)
Cirrhosis		189 (85.1%)
Alcohol consumption		140 (63.1%)
Hepatitis infection B		10 (4.5%)
Hepatitis infection C		31 (14.0%)
Previous treatment for HCC before SIRT		121 (54.5%)
Surgery		41 (18.5%)
Radiofrequency ablation		23 (10.4%)
Trans-arterial chemoembolization (TACE)		59 (26.6%)
I131-lipiodol		21 (9.5%)
Macrovascular invasion		109 (49.0%)
Main portal vein or hepatic vei		21 (9.5%)
Branch portal vein		88 (39.6%)
ALBI score, median (range)		−2.49 (−3.51; −1.17)
ALBI Grade	1	88 (39.6%)
	2	130 (58.6%)
	3	4 (1.8%)
Child−Pugh Score	A5	131 (59%)
	A6	70 (31.5%)
	B7	18 (8.1%)
	B8	3 (1.4%)
Number of liver tumors	1	75 (33.8%)
	2	31 (14.0%)
	3 or more	116 (52.3%)
Median size of the lesion, in mm (range)		67 (10–170)
Median alpha-fetoprotein, median (range)		19 (1–235,700)
Barcelona clinic civer cancer (BCLC) stage	A	1 (0.5%)
	B	67 (30.2%)
	C	154 (68.4%)
	D	0 (0%)
Overall delivered activity, in GBq, median (range)		4.3 (0.23–350)
Dose delivered to tumor, in Gy, median (range)		322.6 (119.3–880)
Lung dose, in Gy, median (range)		4 (0–32)
Nontumoral liver dose, in Gy, median (range)		85.4 (0–221)
Average dose to treated liver, in Gy, median (range)		126.7 (16–252)
First SIRT localization, *n* (%)	Left liver	62 (28%)
	Right liver	151 (68%)
	Bilateral	9 (4%)
Second SIRT, *n* (%)		55 (25%)
Second SIRT localization, *n* (%)	Left liver	27 (12%)
	Right liver	25 (11%)
	Bilateral	3 (11%)
Dose ≥ 205 Gy, *n* (%)		174 (78%)
Glass microspheres, *n* (%)		210 (95%)
Resin microspheres, *n* (%)		12 (5%)

HCC: hepatocellular carcinoma; SIRT: selective internal radioactive therapy.

**Table 2 cancers-13-03794-t002:** Univariate analysis using a Cox model for overall survival.

Variables		Univariate Analysis	
		HR (95% CI)	*p*-Value
Age (continuous)		0.99 (0.98–1.01)	0.54
Gender (male vs. female)		1.07 (0.75–1.72)	0.75
Cirrhosis (yes vs. no)		1.23 (0.83–1.82)	0.31
Etiology of cirrhosis:	Alcohol	1.01 (0.76–1.35)	0.93
	Hepatitis B infection	1.02 (0.52–1.99)	0.95
	Hepatitis C infection	1.43 (0.96–2.12)	0.08
	NASH/others	1.11 (0.84–1.46)	0.46
Treatment before SIRT		0.95 (0.72–1.25)	0.54
—TACE		1.19 (0.97–1.45)	0.09
—Sorafenib		0.92 (0.62–1.34)	0.65
Macrovascular invasion		1.34 (1.01–1.76)	0.04
Bigger tumor size		1.00 (0.99–1.01)	0.16
Performance status (0 vs. 1 or 2)		1.37 (1.02–1.76)	0.16
Child−Pugh score (points)		1.27 (1.04–1.56)	0.02
Child−Pugh score (A vs. B or C)		0.54 (0.34–0.87)	0.01
ALBI score		1.89 (1.39–2.56)	<0.001
ALBI grade (2 or 3 vs. 1)		1.64 (1.23–2.18)	0.001
Multifocal		1.34 (0.99–1.80)	0.05
Baseline alpha-fetoprotein		1 (1–1)	0.03
BCLC (A/B/C/D)		1.47 (1.08–1.99)	0.01
Baseline albumin		0.95 (0.92–0.97)	<0.001
Baseline bilirubin		1.01 (1–1.02)	0.11

NASH: nonalcoholic steatohepatitis TACE: transarterial chemoembolization; SIRT: selective internal radioactive therapy; BCLC: Barcelona clinic liver cancer.

## Data Availability

Request for data should be directed to the corresponding author, J.E.

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
