# Peer review of "ALBI Score Is a Strong Predictor of Toxicity Following SIRT for Hepatocellular Carcinoma"

_cancers, 2021, doi:10.3390/cancers13153794_

Round 1

Reviewer 1 Report

This manuscript by Lescure et al. is of great interest due to the novel relevant insights it provides for people working in the field of liver diseases/HCC. Here are my comments/suggestions to improve the quality:

  • KM curves should be colored to better differentiate the displayed groups.
  • In the Introduction part, first sentence, this reference should definitely be added: Cassim S, Raymond VA, Lacoste B, Lapierre P, Bilodeau M. Metabolite profiling identifies a signature of tumorigenicity in hepatocellular carcinoma. Oncotarget. 2018 Jun 1;9(42):26868-26883. doi: 10.18632/oncotarget.25525. PMID: 29928490; PMCID: PMC6003570.
  • I found the limitations part extremely relevant at the end of the manuscript and wanted to state it to the authors.

Author Response

Responses to reviewer #1:

This manuscript by Lescure et al. is of great interest due to the novel relevant insights it provides for people working in the field of liver diseases/HCC. Here are my comments/suggestions to improve the quality:

  • KM curves should be colored to better differentiate the displayed groups.

We added color to the curves.

  • In the Introduction part, first sentence, this reference should definitely be added: Cassim S, Raymond VA, Lacoste B, Lapierre P, Bilodeau M. Metabolite profiling identifies a signature of tumorigenicity in hepatocellular carcinoma. Oncotarget. 2018 Jun 1;9(42):26868-26883. doi: 10.18632/oncotarget.25525. PMID: 29928490; PMCID: PMC6003570.

We thank the reviewer for this recommendation. We read with great interest the article. Albeit very interesting, this article refers to an in vitro study focusing on the glycolytic and hypoxic pathways. We do not see how it could be related to our article. Was there any mistake in the reference? Or could the reviewer explain how we could introduce this reference in our article?

  • I found the limitations part extremely relevant at the end of the manuscript and wanted to state it to the authors.

We thank the reviewer for this kind comment.

Reviewer 2 Report

In the present manuscript, Lescure and colleagues investigated the prognostic role of ALBI score in patient with HCC treated with SIRT. In this retrospective study, the authors observed that ALBI grade is ascoiated to REILD and OS.

The manuscript is timely and well written. Despite there are several limitations (already highlighted by the authors in the discussion section), the manuscrtipt deserve consideration. I have only some minor comments.

1) Material and methods. I suggest to add a flow chart of the study.

2) Results. Lines 143 - 148. I do not understand the point of the analysis; basically, ALBI and CP are both prognostic factors and reflect liver disease severity...

3) Figures 2, 3A and 3B. Were the p values calculated by chi-sqared test fro trend?

Author Response

In the present manuscript, Lescure and colleagues investigated the prognostic role of ALBI score in patient with HCC treated with SIRT. In this retrospective study, the authors observed that ALBI grade is ascoiated to REILD and OS.

The manuscript is timely and well written. Despite there are several limitations (already highlighted by the authors in the discussion section), the manuscrtipt deserve consideration. I have only some minor comments.

  • Material and methods. I suggest to add a flow chart of the study.

We thank the reviewer for this important comment, and added a flow chart as supplementary material Figure S1.

  • Lines 143 - 148. I do not understand the point of the analysis; basically, ALBI and CP are both prognostic factors and reflect liver disease severity...

We agree with the reviewer. The original intention was to say that as in multivariable analysis, the fact that ALBI persists rather than CP suggests that ALBI is a stronger prognostic factor as compared with CP. However, as this was not clear, and as this analysis could not be conclusive, we prefer to delete the statement.

  • Figures 2, 3A and 3B. Were the p values calculated by chi-sqared test fro trend?

Yes, this was indeed the case. We added the information in the legend.